# QTc Interval Prolongation as an Adverse Event of Azole Antifungal Drugs: Case Report and Literature Review

**DOI:** 10.3390/microorganisms12081619

**Published:** 2024-08-08

**Authors:** Shiori Kitaya, Makoto Nakano, Yukio Katori, Satoshi Yasuda, Hajime Kanamori

**Affiliations:** 1Department of Infectious Diseases, Internal Medicine, Tohoku University Graduate School of Medicine, Sendai 980-8575, Japan; 2Department of Otolaryngology, Head and Neck Surgery, Tohoku University Graduate School of Medicine, Sendai 980-8574, Japan; yukio.katori.d1@tohoku.ac.jp; 3Laboratory Medicine, Department of Infectious Diseases, Kanazawa University, Kanazawa 920-8641, Ishikawa, Japan; 4Department of Cardiovascular Medicine, Tohoku University Graduate School of Medicine, Sendai 980-8574, Japan; vdm@cardio.med.tohoku.ac.jp (M.N.); satoshi.yasuda.c8@tohoku.ac.jp (S.Y.)

**Keywords:** QTc prolongation, long QT syndrome, torsade de pointes, azole antifungal drugs

## Abstract

QTc prolongation and torsade de pointes (TdP) are significant adverse events linked to azole antifungals. Reports on QTc interval prolongation caused by these agents are limited. In this study, we report a case of a 77-year-old male with cardiovascular disease who experienced QTc prolongation and subsequent TdP while being treated with fluconazole for *Candida albicans*-induced knee arthritis. Additionally, a literature review was conducted on cases where QTc prolongation and TdP were triggered as adverse events of azole antifungal drugs. The case study detailed the patient’s experience, whereas the literature review analyzed cases from May 1997 to February 2023, focusing on patient demographics, underlying diseases, antifungal regimens, concurrent medications, QTc changes, and outcomes. The review identified 16 cases, mainly in younger individuals (median age of 29) and women (75%). Fluconazole (63%) and voriconazole (37%) were the most common agents. Concurrent medications were present in 75% of cases, and TdP occurred in 81%. Management typically involved discontinuing or switching antifungals and correcting electrolytes, with all patients surviving. Risk assessment and concurrent medication review are essential before starting azole therapy. High-risk patients require careful electrocardiogram monitoring to prevent arrhythmias. Remote monitoring may enhance safety for patients with implanted devices. Further studies are needed to understand risk factors and management strategies.

## 1. Introduction

Prolongation of the QT interval corrected for heart rate (QTc interval) can potentially lead to life-threatening arrhythmias, such as polymorphic ventricular tachycardia (torsade de pointes, TdP) and ventricular fibrillation [1]. A prolonged QTc interval is classified as either congenital or acquired [1]. Congenital long QT syndrome (LQTS) is defined by the occurrence of genetic mutations affecting sodium, potassium, and calcium channels and is associated with Andersen syndrome, Romano–Ward syndrome, Jervell and Lange-Nielsen syndrome, and Timothy syndrome [2]. The most common causes of acquired QTc interval prolongation are drug use and electrolyte abnormalities, with many drugs (antibiotics, antifungal drugs, antiarrhythmic drugs, antipsychotics, antidepressants, antihistamines, and serotoninergic agents, such as triptans) acting on the important potassium channel subtype IKr, causing QTc interval prolongation [3,4]. QTc prolongation is described as a QTc interval extending beyond 450 ms in males and 470 ms in females, or an absolute increase of more than 60 ms from baseline [5].

Azole antifungal drugs exert their antifungal effects by affecting ergosterol synthesis, which is specific to fungal cell membranes and mainly includes fluconazole, itraconazole, voriconazole, and miconazole [6,7]. Azole antifungal drugs are used worldwide as first-line drugs for the prevention and treatment of invasive fungal infections [8]. However, long-term use has been reported to cause hepatotoxicity and hormone-related side effects, such as gynecomastia, hair loss, decreased libido, oligospermia, azoospermia, erectile dysfunction, hypokalemia, hyponatremia, and adrenal insufficiency [8].

Reports of cardiovascular adverse events such as TdP and QTc prolongation associated with azole antifungal drugs have been increasing, with an incidence reported to occur in approximately 4.7–39.1% of cases [9,10,11,12,13,14]. These drugs may be involved in the elevation of plasma concentrations of QTc-prolonging drugs via the same metabolic pathway, owing to the inhibition of multiple cytochrome P450 enzymes in the liver and gastrointestinal tract [15,16,17]. In animal experiments using long-eared white rabbits, it has also been shown that fluconazole may prolong the QTc interval and exert proarrhythmic activity by inhibiting human ether-a-go-go-related gene (hERG) protein trafficking [18]. Additionally, it has been reported that ventricular arrhythmias can occur even in the absence of other QTc-prolonging drugs [19].

Reports on QTc interval prolongation caused by azole antifungal drugs are mainly case reports, and studies reporting their characteristics and associated risk factors are limited. In this study, we report a case of QTc interval prolongation due to adverse effects of azole antifungal drugs, with subsequent TdP treatment using an implantable cardioverter-defibrillator with biventricular pacing function. Additionally, we conducted a literature review to elucidate the clinical features of QTc prolongation associated with azole antifungal drugs.

## 2. Case Report

A 77-year-old male was admitted to our hospital after multiple activations of his cardiac resynchronization therapy-defibrillator (CRT-D). The patient had undergone bilateral knee arthroplasty 8 years previously, followed by a diagnosis of chronic atrial fibrillation and heart failure, for which he had undergone CRT-D implantation 3 years previously. Additionally, he had a history of transient ischemic attack 25 years previously and had undergone percutaneous coronary intervention for acute myocardial infarction 10 years previously prior to presenting to our hospital. During a reoperation for bilateral knee arthroplasty 2 months prior, *Candida albicans* was detected in the joint fluid, and the patient was treated for one month with oral fluconazole 200 mg twice daily for *Candida* knee arthritis. The patient was on regular medications including digitalis, pimobendan, and azosemide for heart failure, amiodarone and edoxaban tosilate hydrate for chronic atrial fibrillation, edoxaban tosilate hydrate for old myocardial infarction, alogliptin and pioglitazone for diabetes mellitus, rosuvastatin for hyperlipidemia, silodosin for urinary dysfunction associated with benign prostatic hyperplasia, acetaminophen for knee pain, esomeprazole for gastroesophageal reflux disease, trazodone and verospiron for depression, suvorexant for insomnia, and memantine for Alzheimer’s disease.

Upon admission to our hospital, vital signs were normal, and no signs of infection were observed at the CRT-D implantation site. The 12-lead electrocardiogram (ECG) at admission showed a QTc interval of 490 ms, which was prolonged to 536 ms the following day (markedly prolonged compared to 435 ms 3 months earlier) (Figure 1). Transthoracic echocardiography revealed a left ventricular ejection fraction of 25–30%, indicating severe diffuse impairment. Transesophageal echocardiography revealed no evident vegetation on the leads of the CRT-D, atrial or tricuspid valves, or the aortic valve. The left knee joint was swollen, raising suspicion of recurrent *Candida* arthritis, but serum β-D-glucan was negative, and it was diagnosed as pseudogout by an orthopedic surgeon. The potassium level was at the lower limit of normal, at 3.5 mg/dL (against the reference ranges of 3.5–5 mg/dL), and liver and kidney function were normal. The blood culture results obtained on the day of admission were negative.

Eight hours after admission, premature ventricular contractions occurred, which led to TdP (Figure 2). The patient experienced a transient loss of consciousness but regained consciousness after CRT-D shock. The patient was taking both amiodarone and fluconazole at the time of admission. Amiodarone is a class III antiarrhythmic agent that prolongs the QTc interval by antagonizing IKr and IKs and repolarizing potassium currents, resulting in an increased risk of TdP. Amiodarone is metabolized by the cytochrome P450 enzyme CYP3A4 to N-desethylamiodarone, and fluconazole is believed to potentiate QTc interval prolongation caused by amiodarone by inhibiting this metabolism [20]. Additionally, fluconazole has been reported to cause acquired LQTS by directly inhibiting the hERG potassium channels and interfering with hERG protein trafficking [21]. Based on these findings, fluconazole, amiodarone, and hypokalemia were considered the causes of QTc prolongation in this case. Consultation with the Infectious Disease Department led to a change in the antifungal medication from fluconazole to micafungin (100 mg once daily). Intravenous potassium chloride, magnesium sulfate, and oral potassium L-aspartate were administered for electrolyte correction. After the discontinuation of fluconazole on day 12, the patient’s QTc interval remained below 450 ms, and there were no further episodes of life-threatening ventricular arrhythmias. As the battery of the CRT-D device was low and the blood cultures remained negative, the device was replaced on day 15 of hospitalization. Potassium levels gradually normalized and stabilized within the normal range on day 17 of hospitalization. The patient recovered without issues and was transferred to a nearby hospital on day 31 of hospitalization while continuing micafungin intravenous therapy. Subsequent follow-ups over 2 years and 4 months did not reveal any recurrence of QTc prolongation.

## 3. Literature Review Results

### 3.1. Literature Search and Selection Criteria

A search strategy, including specific databases and search terms, was identified and agreed upon by the two authors. We searched the published literature (from May 1997 to February 2023) via PubMed using the following keywords: “Long QT Syndrome” AND (“Econazole” OR “Miconazole” OR “Fluconazole” OR “Fosfluconazole” OR “Itraconazole” OR “Voriconazole”). Forty references were carefully reviewed. A summary of articles identified in the initial search was independently screened to determine whether they met the inclusion criteria. The full texts were reviewed for relevant and ambiguous articles. All articles that met the inclusion criteria were identified, and the full text of each article was searched for additional relevant studies. Each article was assigned a numerical code for organizational identification. Finally, 15 studies [2,19,22,23,24,25,26,27,28,29,30,31,32,33] were selected for the analysis. This includes all reported QTc prolongation syndrome cases believed to be caused by antifungal drugs. Publications not published in English, review articles, and reports that did not include the occurrence of QTc prolongation syndrome in humans were excluded. This study was approved by the Human Ethics and Clinical Trial Committee of Tohoku University Hospital (approval no. 2019-1-270).

### 3.2. Data Extraction

The following variables were extracted: the author; year of publication; patient characteristics (e.g., age, sex); underlying disease; pre-existing cardiovascular disease; organisms; fungal infection focus; regimen and dosages of antifungal drugs; antifungal drug administration purposes (therapeutic, prophylactic); use of other medications potentially causing QTc prolongation besides antifungal drugs; serum electrolyte levels; interval between antifungal drug administration and QTc prolongation detection; longest QTc interval; TdP presence; other cardiovascular events besides TdP; QTc interval change after episode occurrence; treatment (management of antifungal drug post-event, and QTc prolongation and cardiovascular event treatment); presence or absence of disease relapse; and outcome.

### 3.3. Evaluation of the QTc Prolongation Risk for Each Medication

In this review, the risk of QTc prolongation for each drug was assessed based on the previously reported literature [34] and the QT drug list from CredibleMeds [35]. Drugs were categorized as follows: List 1 included drugs with a known risk of TdP; List 2 included drugs with a possible risk of TdP; List 3 included drugs with a conditional risk of TdP; and List 4 included drugs to be avoided by patients with congenital LQTS. These lists are evidence-based and continuously updated with new information.

### 3.4. Statistical Analysis

Fisher’s exact test was used to compare the proportions of categorical variables between the two groups. The analysis was performed using the JMP Pro 16 statistical analysis software (SAS Institute, Cary, NC, USA, 2021). Differences were considered significant if the *p*-value was under 0.05.

### 3.5. Review Results

A case series of QTc prolongation due to azole antifungals was revealed through a literature review of 16 cases (from 15 articles) [2,19,22,23,24,25,26,27,28,29,30,31,32,33]. The results of the literature review and cases from our institution are summarized in Table 1, and detailed information is provided in Appendix A.

QTc prolongation caused by azole antifungals predominantly occurred in younger individuals (median age of 29 years, interquartile range (IQR) of 15.8–45). There was a tendency for more females than males to be affected (75% vs. 25%). While various underlying conditions were present, relatively higher occurrences were noted in hematologic malignancies, such as acute myeloid leukemia, recurrent acute lymphoblastic leukemia, and mediastinal B-cell lymphoma (4/16 cases, 25%). Only seven patients (44%) had a history of cardiovascular disease.

Antifungals were administered for therapeutic purposes and accounted for 10/16 cases (63%). In contrast, in 6/16 cases (37%), antifungal drugs were administered prophylactically. The most frequently administered antifungal drug was fluconazole (10/16 cases, 63%), followed by voriconazole (6/16 cases, 37%), and liposomal amphotericin B (4/16 cases, 25%). A single antifungal administration before QTc prolongation occurred in 10/16 cases (63%). Conversely, simultaneous administration of multiple antifungals was observed in 4/16 cases (25%), and sequential administration of different antifungals over time was noted in 3/16 cases (19%). The median duration from antifungal administration to QTc prolongation was 5 days (IQR 4.5–11). The median duration of the longest QTc interval was 551 ms (IQR 515–590 ms). Other medications potentially contributing to QTc prolongation were concurrently administered; they accounted for 12/16 cases (75%). The most common type of medication was antibiotics, present in 6/12 cases (50%). Antidepressants and proton pump inhibitors were the next most common, present in 4/12 cases each (33%).

TdP occurred in 13/16 cases (81%) with QTc prolongation. Other concurrent cardiovascular events included ventricular fibrillation, cardiac arrest, and syncope in 2/16 cases each (13%), and pulseless monomorphic ventricular tachycardia, ventricular bigeminy, and trigeminy beats in 1/16 cases each (6%). Among the patients with QTc prolongation, antifungal discontinuation was observed in 7/16 (44%), switching to non-azole antifungal drugs in 3/16 (19%), once discontinued and then resumed after improvement of the QTc interval in 3/16 (19%), switching to other azole antifungal drugs in 1/16 (6%), dosage reduction in 1/16 (6%), and unknown in 1/16 (6%). The median duration until the QTc interval returned to within the normal range after treatment for QTc prolongation was 7 days (IQR 3.5–18). Recurrence of QTc prolongation after treatment was observed in 3/16 cases (19%), with two cases attributed to fluconazole or voriconazole re-administration. Another case was of unknown cause and subsequently required the implantation of a cardioverter-defibrillator. All patients in this review survived even after QTc prolongation.

## 4. Discussion

### 4.1. Age and Underlying Diseases

In this case, QTc prolongation occurred in a 77-year-old elderly male patient; however, in this review, a higher incidence of QTc prolongation after the use of azole antifungal drugs was observed in younger patients (median age of 29 years, IQR of 15.8–45). In terms of the sex ratio, a similar trend of higher prevalence in females, as reported previously, was also observed in this review. Females inherently have longer QTc intervals than males [36]. Due to differences in specific cardiac ion densities, females are predisposed to QTc prolongation and associated arrhythmias, making them more susceptible to developing TdP when exposed to drugs that further prolong QTc intervals [29,37]. The possibility of selection bias, owing to frequently reported case studies of QTc prolongation in younger individuals, who are typically considered to be at low risk, cannot be completely ruled out. However, based on the results of this review, it can be inferred that the risk of events such as QTc prolongation during azole antifungal therapy and subsequent TdP is higher in younger individuals and females.

The patient in this case had multiple cardiovascular diseases, including chronic atrial fibrillation, heart failure, and an old myocardial infarction. In contrast, in this review, many patients without pre-existing heart disease were observed (9/16 cases, 56%). Therefore, even in patients without a history of heart disease, attention should be paid to QTc prolongation during azole antifungal drug administration.

Among the underlying diseases in the patients reviewed, hematological malignancies were relatively common (4/16, 25%). Patients with hematologic malignancies are predisposed to QTc prolongation owing to factors such as hypokalemia and/or hypomagnesemia resulting from chemotherapy-induced diarrhea, renal toxicity, and multiple drug therapies [9]. In this review, all patients with a history of hematological malignancies were found to be receiving azole antifungal drugs in combination with other medications known to prolong the QTc interval (piperacillin–tazobactam, imatinib mesylate, levofloxacin, and dasatinib). Additionally, appropriate clinical and ECG monitoring is recommended for patients with hematologic malignancies or those undergoing hematopoietic stem cell transplantation and receiving combined fluoroquinolone-azole therapy [11]. Considering the results of this review, it is necessary to conduct more careful ECG monitoring during treatment, as patients with hematologic malignancies have the characteristics of being predisposed to QTc prolongation and are often receiving a combination of azole antifungal drugs and other QTc-prolonging drugs.

### 4.2. Relationship between QTc Interval Prolongation and TdP, along with Risk Factors

QTc prolongation refers to the extension of the QTc interval beyond 450 ms in males and 470 ms in females (or an absolute increase > 60 ms from baseline) [5]. For every 10 ms increase in the QTc interval, the risk of TdP is estimated to increase by 5–7%, and QTc intervals exceeding 500 ms are significantly associated with TdP onset [4]. According to the American Heart Association/American College of Cardiology consensus statement on TdP prevention, an immediate response involving risk factor modification or alternative drug therapy is recommended upon identifying QTc intervals exceeding 500 ms or showing an increase of at least 60 ms from baseline [38]. However, there are no clear criteria for the degree of QTc interval prolongation or the occurrence of TdP. In this review, because cases leading to TdP were observed even with QTc prolongation of approximately 490 ms [26], it is important to be mindful of the potential for TdP, even with mild QTc prolongation.

Risk factors for TdP in hospitalized patients include QTc intervals exceeding 500 ms or an increase of 60 ms from the baseline, congenital LQTS, use of multiple QTc-prolonging drugs, rapid administration of QTc-prolonging drugs, history of drug-induced TdP, heart diseases (congestive heart failure and myocardial infarction), advanced age, female sex, electrolyte abnormalities (hypokalemia, hypomagnesemia, hypocalcemia, etc.), administration of diuretics, chronic kidney/hepatic failure, bradycardia, and polymorphisms in genes encoding ion channels [39,40]. When using azole antifungal drugs, it is essential to assess whether these risk factors are present beforehand and accordingly conduct a risk assessment for the patient.

### 4.3. Electrolyte Abnormalities

Electrolyte abnormalities are considered important modifiable risk factors for QTc prolongation [38]. Hypokalemia may decrease the effectiveness of the delayed rectifier potassium channel in the cardiac tissue and increase drug binding to the channel, potentially leading to excessive repolarization prolongation [41]. Hypomagnesemia effectively increases the amplitude of early depolarization, which can cause TdP [39]. In this case, the serum potassium level at the time of the event was at the lower end of the normal range, and hypokalemia or hypomagnesemia was observed in 8/16 cases (50%). In these cases, besides QTc prolongation induced by the use of azole antifungal drugs, further prolongation of QTc intervals may have been exacerbated by the aforementioned mechanisms related to electrolyte abnormalities. The recommended strategies for preventing drug-induced QTc interval prolongation include actively replenishing magnesium to levels > 2 mg/dL and potassium to levels > 4 mEq/L [38]. Therefore, actively correcting these levels to the appropriate ranges during the treatment of patients with hypokalemia or hypomagnesemia may help reduce the risk of QTc prolongation associated with azole antifungal drug use.

### 4.4. Duration, Dosage, and Purpose of Administration of Azole Antifungal Drugs

In our case, the duration from the initiation of oral azole antifungal therapy to the onset of QTc prolongation was 1 month, whereas in this review, the median duration to onset was 5 days (IQR 4.5–11). The median time to onset of TdP/QTc prolongation was 9 days (IQR 2–11) for voriconazole, 10 days (IQR 2–85) for fluconazole, 8 days (IQR 3–30) for itraconazole, and 15 days (IQR 10–24) for posaconazole [40]. The site and degree of CYP inhibition by azole agents vary depending on the type of azole. However, based on the results of this review, because QTc prolongation can occur relatively early after therapy initiation, careful ECG monitoring should be initiated early after azole antifungal drug administration in high-risk patients, regardless of the type of medication.

The relationship between azole antifungal drug dosage and QTc prolongation remains controversial. An increased incidence of clinically significant QTc changes was observed with increasing doses [11]. This is attributed to higher drug levels enhancing the blockade of delayed rectifier potassium channels in the cardiac tissue and increasing the inhibition of other drug metabolic pathways [11]. In this case, fluconazole was administered orally at a dose of 200 mg every 12 h, approximately four times the usual dose for candidiasis. Although there are no clear criteria regarding the dosage of azole antifungal drugs and the occurrence of QTc prolongation, cautious monitoring is considered to contribute to patient safety in patients at high risk for QTc prolongation when using triazole agents, regardless of the dosage.

Regarding the purpose of antifungal drug administration, in this review, six patients (37%) developed QTc prolongation during prophylactic administration. Similarly, previous studies have reported QTc prolongation cases during prophylactic administration of antifungal drugs [10]. Therefore, it is important to note that QTc prolongation may occur even during prophylactic azole antifungal drug administration and not just for therapeutic purposes.

### 4.5. Other QTc-Prolonging Drugs Aside from Azole Antifungal Drugs

Drug-induced LQTS can be caused by a variety of drugs other than azole antifungal drugs, including antiarrhythmic, antipsychotic, antibiotic, and antiallergic drugs [42]. CredibleMeds publishes a list of drugs on its website associated with the risk of QTc prolongation, with currently over 300 drugs listed [35]. These drugs are classified into four groups based on their risk: List 1 includes drugs with a known risk of TdP; List 2 includes drugs with a possible risk of TdP; List 3 includes drugs with a conditional risk of TdP; and List 4 includes drugs to be avoided by patients with congenital LQTS [35]. Among azole antifungal drugs, fluconazole is classified under List 1, whereas ketoconazole, itraconazole, voriconazole, and posaconazole are classified under List 3. In this case, the patient was taking three QTc-prolonging drugs in addition to fluconazole: amiodarone (List 1), esomeprazole (List 3), and trazodone (List 3). It is plausible that these medications, along with fluconazole, contributed to the QTc prolongation observed in this patient. Additionally, in this review, 75% (12/16 cases) of patients were found to be taking other drugs classified as having a risk of QTc prolongation besides azole antifungal drugs. Therefore, when using azole antifungal drugs, it is necessary to check the patient’s medication history to confirm whether they are taking other drugs with a risk of QTc prolongation. Taking measures, such as switching or discontinuing management and risk assessment of LQTS QTc-prolonging drugs, when possible, is crucial for preventing QTc prolongation.

### 4.6. Regarding QTc Prolongation Management

Regarding QTc prolongation management, in cases where the suspected antifungal drug was discontinued, almost all cases (6/7 cases, 86%) did not show QTc prolongation recurrence (the cause of recurrence in one case remained unclear [24]). It is generally believed that discontinuing the suspected drug can prevent QTc prolongation recurrence; however, it is also important to conduct ECG monitoring to ensure no recurrence thereafter. In recent years, the market for mobile cardiac telemetry devices has expanded rapidly [43]. This system allows medical institutions to monitor cardiac device information online, enabling prompt therapeutic interventions when necessary [44]. In this case, ECG information from the CRT-D during defibrillation for TdP was transmitted to our hospital, which enabled prompt hospitalization and therapeutic intervention. The use of such remote monitoring systems by patients with implantable devices may improve patient outcomes.

Despite dosing adjustments based on body weight, the pharmacokinetics of azole antifungal drugs exhibit considerable inter-individual variability due to factors such as hepatic function, cytochrome P450 polymorphisms, concomitant medications, drug interactions, and clinical symptoms, which may lead to abnormal trough concentrations [45,46,47,48]. Therapeutic drug monitoring (TDM) of itraconazole, voriconazole, or posaconazole may minimize the toxicity associated with excessive antifungal drug concentrations and improve patient outcomes by preventing treatment failure due to therapeutic insufficiency [49,50]. The Infectious Diseases Society of America recommends TDM for itraconazole, voriconazole, or posaconazole in the treatment of invasive fungal infections [51,52,53,54,55], particularly in pediatric patients, critically ill patients, those who are obese, and those with hepatic or renal impairment [56]. Data regarding the relationship between the blood concentrations of azole antifungal drugs and the QTc prolongation extent are currently limited. However, TDM may help to maintain appropriate blood concentrations of azole antifungal drugs, potentially contributing to QTc prolongation and TdP prevention.

Furthermore, a risk score has been proposed to identify patients at high or low risk of QTc prolongation [34]. In an observational study targeting patients starting with haloperidol or QTc-prolonging antibiotics/antifungals, the performance characteristics of the preliminary risk score (RISQ-PATH score) were calculated based on a systematic review of risk factors (receiver operating characteristic analysis determined a score of less than 10 points as low risk for QTc prolongation). The RISQ-PATH score had a negative predictive value of 98% and could exclude low-risk patients, showing promise in excluding patients from further follow-up when starting QTc-prolonging drugs. The RISQ-PATH score in the cases we encountered was 15.5, indicating an intermediate or higher risk, thus necessitating careful ECG monitoring upon QTc-prolonging drug initiation. Pre-assessment using the RISQ-PATH score when initiating QTc-prolonging drugs and conducting more thorough ECG monitoring in high-risk patients classified as high-risk may serve as useful criteria for follow-up decisions.

Furthermore, chemical modifications and the development of new formulations, including azole antifungal drugs, are being conducted to improve therapeutic efficacy and reduce adverse events. Nano-structured systems have been reported to act as carriers for antifungal drugs, potentially enhancing drug bioavailability and reducing toxicity [57]. These studies are expected to contribute to the reduction of adverse events associated with antifungal drugs, including azole antifungal drugs.

## 5. Conclusions

In this report, we describe a case of TdP triggered by QTc prolongation while taking oral fluconazole to treat knee arthritis caused by *C. albicans*. The remote monitoring of the patient utilizing implantable devices, rapid detection of QTc prolongation onset, and subsequent treatment led to a favorable prognosis. Furthermore, based on the information in this review, we concluded the following regarding QTc prolongation associated with the use of azole antifungal drugs: (1) There is a higher risk of QTc prolongation and subsequent events, such as TdP, in young individuals and females. (2) Even in patients without a history of heart disease, attention should be paid to QTc prolongation. (3) It is important to assess the risk of TdP and check for the concurrent use of other drugs that pose a risk of QTc prolongation prior to the use of azole antifungal drugs. Additionally, measures, such as substituting or discontinuing medications that pose a risk of QTc prolongation, if possible, should be undertaken to prevent QTc prolongation. (4) For high-risk patients with QTc prolongation receiving azole antifungal drugs, regardless of the type or dosage of medication, careful ECG monitoring should commence immediately after administration. Utilizing remote monitoring systems, especially for patients with implantable devices, may contribute to improved patient outcomes.

## Figures and Tables

**Figure 1 microorganisms-12-01619-f001:**
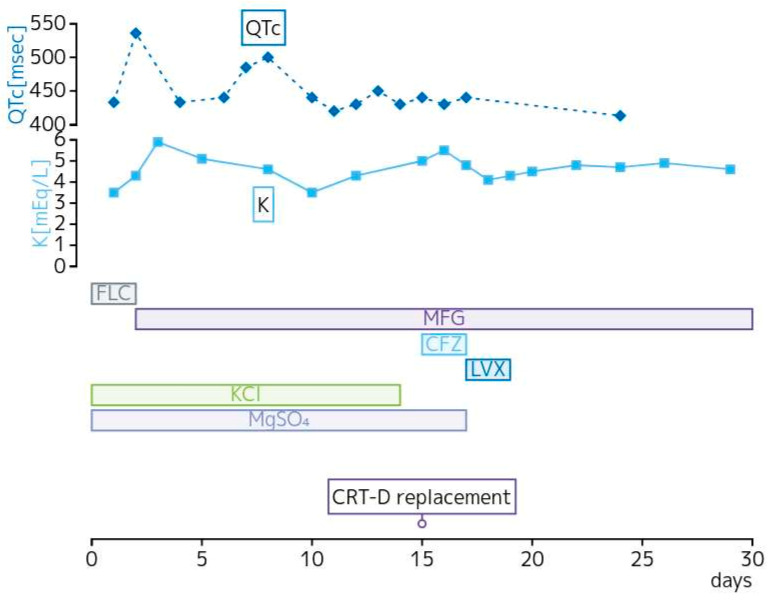
Changes in QTc interval, serum potassium concentration, and treatment course during hospitalization. CFZ, cefazolin; CRT-D, cardiac resynchronization therapy-defibrillator; FLC, fluconazole; K, potassium; KCl, potassium chloride; LVX, levofloxacin; MFG, micafungin; MgSO4, magnesium sulfate; QTc, QT interval corrected for heart rate.

**Figure 2 microorganisms-12-01619-f002:**
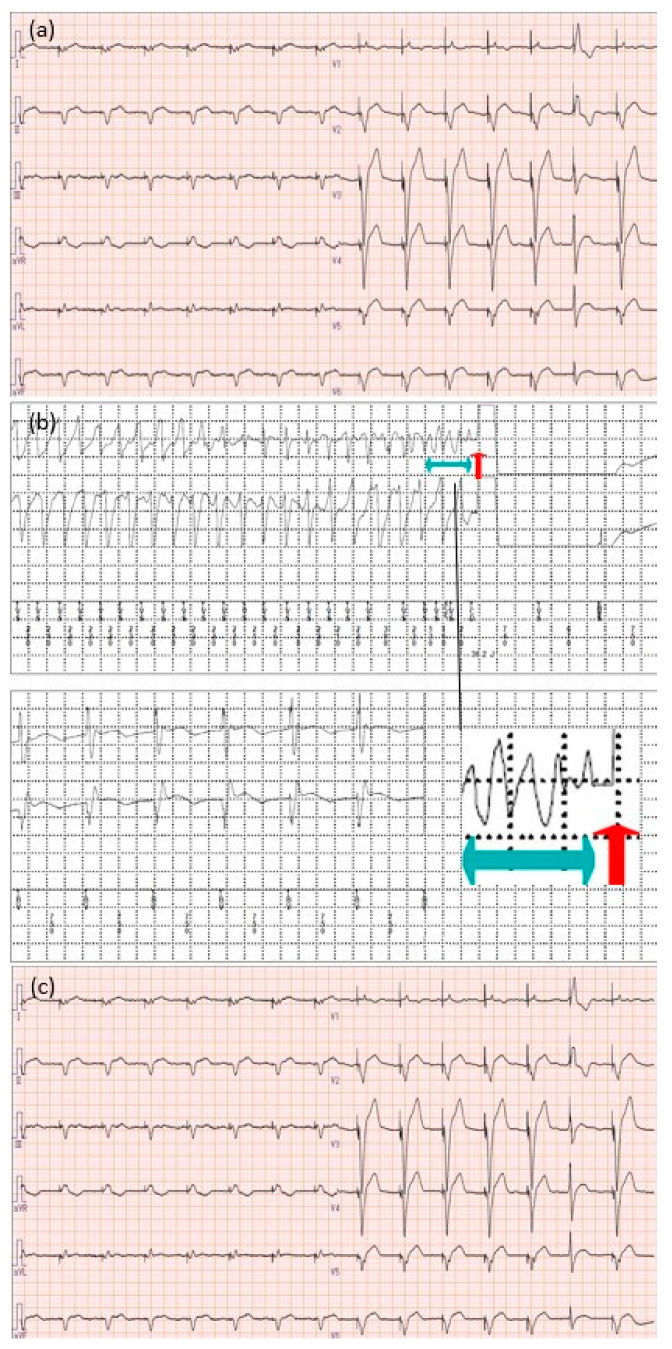
Electrocardiogram findings (**a**) at admission; (**b**) occurrence of torsade de pointes and subsequent defibrillation using a cardiac resynchronization therapy-defibrillator; (**c**) at discharge. The light blue arrow indicates the area of the electrocardiogram where the torsade de pointes occurred, and the red arrow indicates the defibrillation activation timing. Torsade de pointes is a specific form of polymorphic ventricular tachycardia observed in patients with QTc prolongation. It is characterized by rapid and irregular QRS complexes that appear to twist around the baseline on the electrocardiogram.

**Table 1 microorganisms-12-01619-t001:** Characteristics of cases of QTc prolongation caused by azole antifungal drugs.

Case	First Author, Year, Reference	Age (Years), Sex	Underlying Disease	Preexisting Cardiovascular Disease	Antifungal Drug Regimen	Antifungal Drug	Antifungal Drug Administration Purpose	Other Drugs Potentially Causing QTc Prolongation	Serum Electrolyte Levels	Interval between Antifungal Drug Administration and QTc Prolongation Detection	Longest QTc Interval	TdP Presence	Management of Antifungal Drug Post-Event	Disease Relapse	Outcome
1	Wassmann, 1999 [19]	59, F	Liver cirrhosis and peritonitis	None	Single	Fluconazole	Therapeutic	None	Normal	Unknown	600 ms	Yes	Discontinuation	No	Survived
2	Dorsey, 2000 [22]	57, F	Mild diabetes mellitus, chronic depression, remote deep venous thrombosis with pulmonary embolism, and chronic renal insufficiency	Hypertension	Single	Fluconazole	Therapeutic	Amitriptyline (List 3) and sertraline (List 3)	3.1 mEq/L K	Unknown	Unknown	Yes	Reduction	No	Survived
3	Hinterseer, 2006 [23]	25, F	Acute respiratory distress syndrome, septic shock, and multiple organ failure	Dilated left ventricle with severely impaired left ventricular function	Single	Fluconazole (empirical administration)	Prophylactic	Erythromycin (List 1) and norepinephrine (List 4)	Normal K level; 0.5 mg/dL Mg	Unknown	510 ms	Yes	Unknown	No	Survived
4	Pham, 2006 [24]	33, F	Systemic lupus erythematosus, hemolytic anemia, pulmonary eosinophilia, constrictive pericarditis, and proliferative glomerulonephritis	Hypertension	Single	Fluconazole	Therapeutic	Albuterol (List 4), citalopram (List 1), furosemide (List 3), isoproterenol (List 4), and pantoprazole (List 3)	Normal	5 days	675 ms	Yes	Discontinued and then resumed after improvement of the QTc interval	Yes (6 weeks later, once, after fluconazole re-administration)	Survived
5	Eiden, 2007 [25]	14, F	Acute myeloid leukemia and neutropenia	None	Multiple (simultaneously and consecutively)	Liposomal amphotericin B to caspofungin, voriconazole to posaconazole	Therapeutic	Piperacillin/tazobactam (List 3)	Normal K level; 0.4 mg/dL Mg	5 days after voriconazole initiation	500 ms	Yes	Discontinuation	Yes (3 months later, once)	Survived
6	Esch, 2008 [26]	11, M	Neurofibromatosis-1, paraplegia, and neurogenic bowel/bladder dysfunction	None	Single	Fluconazole	Therapeutic	Furosemide (List 3) and chloral hydrate (List 3)	3.1 mEq/L K; normal Mg level	5 days	490 ms	Yes	Discontinuation	No	Survived
7	Tacken, 2011 [27]	69, F	Tubo-ovarian abscess and pre-sacral abscess	Atrial fibrillation and hypertension	Single	Fluconazole	Prophylactic	Piperacillin/tazobactam (List 3), esomeprazole (List 3), propofol (List 1), and sevoflurane (List 1)	Normal	2 days	626 ms	Yes	Switch to caspofungin	No	Survived
8	Aypar, 2011 [28]	15, M	―	Ventricular septal defect	Multiple (simultaneously)	Liposomal amphotericin B and voriconazole	Therapeutic	None	Normal	During initial voriconazole administration	500 ms	Yes	Switch to caspofungin	No	Survived
9	Aypar, 2011 [28]	12, F	Relapsed acute lymphoblastic leukemia	None	Multiple (simultaneously)	Voriconazole and caspofungin	Therapeutic	Imatinib mesylate (List 2)	2.8 mEq/L K; 1.4 mg/dL Mg	11 days	570 ms	No	Discontinued and then resumed after improvement of the QTc interval	No	Survived
10	Elbey, 2012 [29]	34, F	Nephrotic syndrome	Rheumatic mitral valve disease	Multiple (consecutively)	Amphotericin B to voriconazole	Prophylactic	Piperacillin/tazobactam (List 3)	Normal K level; 1.2 mg/dL Mg	4 days after voriconazole initiation	580 ms	Yes	Discontinued and then resumed after improvement of the QTc interval	Yes (several days later, once, after voliconazole re-administration)	Survived
11	Panos, 2016 [2]	26, F	Ectopic calcifications in the left hip joint, surgical site infection, and sepsis	None	Multiple (consecutively)	Liposomal amphotericin B, voriconazole, and oral posaconazole (empirical administration)	Prophylactic	None	Normal	11 days after voriconazole initiation	540 ms	Yes	Discontinuation	No	Survived
12	Trang, 2017 [30]	22, F	Advanced cystic fibrosis	None	Single	Voriconazole	Therapeutic	Ciprofloxacin (List 1) and azithromycin (List 1)	Normal	QTc prolongation present prior to antifungal drug initiation	613 ms	No	Switch to isavuconazole	No	Survived
13	Tilton, 2019 [31]	32, F	Mediastinal B-cell lymphoma and neutropenia	None	Single	Fluconazole	Prophylactic	Levofloxacin (List 1)	3.4 mEq/L K; normal Mg level	Unknown	551 ms	No	Discontinuation	No	Survived
14	Yüksekgönül, 2021 [32]	16, F	Diffuse axonal injury and pneumonia	None	Single	Fluconazole	Prophylactic	None	Unknown	6 days	560 ms	Yes	Discontinuation	No	Survived
15	Yuan, 2023 [33]	41, M	Acute myeloid leukemia	None	Multiple (simultaneously)	Micafungin and fluconazole	Therapeutic	Dasatinib (List 2)	3.2 mEq/L K; 0.9 mg/dL Mg	25 days	520 ms	Yes	Discontinuation	No	Survived
16	Present case, 2023	77, M	―	Chronic atrial fibrillation, heart failure, and old myocardial infarction	Single	Fluconazole	Therapeutic	Amiodarone (List 1), esomeprazole (List 3), and trazodone (List 3)	Normal	1 month	536 ms	Yes	Switch to micafungin	No	Survived

In this review, the assessment of the risk of QTc prolongation for each drug was conducted by referencing the QT drugs list from CredibleMeds [32]. Drugs were categorized into the following: List 1, drugs with a known risk for TdP; List 2, drugs with a possible risk for TdP; List 3, drugs with a conditional risk for TdP; and List 4, drugs to be avoided by patients with congenital long QT syndrome. Serum electrolyte levels indicate the values at the time of QTc prolongation events. K and Mg levels are considered normal if they are within the ranges of 3.5–5 mg/dL and 1.7–2.6 mg/dL, respectively. Abbreviations: F, female; K, potassium; M, male; Mg, magnesium; TdP, torsade de pointes.

## Data Availability

The data underlying this article cannot be publicly shared to protect the privacy of the individuals who participated in the study.

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
