# Peer review of "QTc Interval Prolongation as an Adverse Event of Azole Antifungal Drugs: Case Report and Literature Review"

_microorganisms, 2024, doi:10.3390/microorganisms12081619_

Round 1

Reviewer 1 Report

Comments and Suggestions for Authors

The review titled "QTc Interval Prolongation as an Adverse Event of Azole Antifungal Drugs: Case Report and Literature Review" by Shiori Kitaya et al. reported a case of a 77-year-old male with cardiovascular disease who experienced QTc prolongation and subsequent TdP while being treated with fluconazole for Candida albicans-induced knee arthritis. Additionally, a literature review was conducted on cases where QTc prolongation and TdP were triggered as adverse events of azole antifungal drugs. The case highlights the risks associated with azole antifungal use, especially in patients with pre-existing conditions. It effectively illustrates the complexities of managing such adverse events. This review focuses on QTc interval prolongation as an adverse event of azole antifungal drugs. The manuscript is comprehensive and well-structured, addressing the following comments will enhance its clarity, impact, and overall quality.

1. The introduction effectively outlines the clinical significance of QTc prolongation and its potential consequences. However, it could benefit from a brief mention of the prevalence of azole antifungal use and their general safety profile.

2. The literature review is thorough, covering relevant studies on QTc prolongation linked to azole antifungals. However, the discussion could benefit from a more critical analysis of the data, particularly regarding differences in study methodologies and their impact on findings.

3. The discussion appropriately interprets the case report and literature review findings. It provides valuable insights into the risk factors and management strategies for QTc prolongation due to azole antifungals. The authors offer practical recommendations for clinicians, such as careful medication review and ECG monitoring. However, the discussion could be enhanced by suggesting specific protocols or guidelines for monitoring and management.

4. he conclusion effectively summarizes the key findings and clinical implications. However, it could be strengthened by outlining specific areas for future research, particularly regarding the mechanisms of drug interactions and the development of safer antifungal therapies.

Author Response

Reviewer 1

The review titled "QTc Interval Prolongation as an Adverse Event of Azole Antifungal Drugs: Case Report and Literature Review" by Shiori Kitaya et al. reported a case of a 77-year-old male with cardiovascular disease who experienced QTc prolongation and subsequent TdP while being treated with fluconazole for Candida albicans-induced knee arthritis. Additionally, a literature review was conducted on cases where QTc prolongation and TdP were triggered as adverse events of azole antifungal drugs. The case highlights the risks associated with azole antifungal use, especially in patients with pre-existing conditions. It effectively illustrates the complexities of managing such adverse events. This review focuses on QTc interval prolongation as an adverse event of azole antifungal drugs. The manuscript is comprehensive and well-structured, addressing the following comments will enhance its clarity, impact, and overall quality.

  1. The introduction effectively outlines the clinical significance of QTc prolongation and its potential consequences. However, it could benefit from a brief mention of the prevalence of azole antifungal use and their general safety profile.

Response: We appreciate the reviewer’s comment. Following the reviewer's suggestion, we have added a description of the frequency of use of azole antifungals for Candida infections and their common side effects to the introduction. Although we were unable to find a specific and detailed percentage regarding the frequency of use of azole antifungals in the literature we reviewed, we have cited references as thoroughly as possible.

“Azole antifungal drugs are used worldwide as first-line drugs for the prevention and treatment of invasive fungal infections [8]. However, long-term use has been reported to cause hepatotoxicity and hormone-related side effects, such as gynecomastia, hair loss, decreased libido, oligospermia, azoospermia, erectile dysfunction, hypokalemia, hyponatremia, and adrenal insufficiency [8].” (Please see Page 2, Lines 50–54)

  1. The literature review is thorough, covering relevant studies on QTc prolongation linked to azole antifungals. However, the discussion could benefit from a more critical analysis of the data, particularly regarding differences in study methodologies and their impact on findings.

Response: We are grateful for the reviewer’s input. Currently, various studies related to QTc prolongation associated with azole antifungal agents are being conducted. However, this review focuses exclusively on case reports. Therefore, it can be assumed that the research methodologies are fundamentally the same. Traditionally, QTc prolongation associated with azole antifungal drugs has been considered more common in elderly patients. However, this review has revealed that it is also frequently observed in younger patients, which is clinically very significant. Accordingly, we have provided a detailed discussion on this point in the discussion section.

“In this case, QTc prolongation occurred in a 77-year-old elderly male patient; however, in this review, a higher incidence of QTc prolongation after the use of azole antifungal drugs was observed in younger patients (median age 29 years, IQR 15.8–45). In terms of the sex ratio, a similar trend of higher prevalence in females, as reported previously, was also observed in this review. Females inherently have longer QTc intervals than males [36]. Due to differences in specific cardiac ion densities, females are predisposed to QTc prolongation and associated arrhythmias, making them more susceptible to developing TdP when exposed to drugs that further prolong QTc intervals [29,37]. The possibility of selection bias, owing to frequently reported case studies of QTc prolongation in younger individuals, who are typically considered to be at low risk, cannot be completely ruled out. However, based on the results of this review, it can be inferred that the risk of events such as QTc prolongation during azole antifungal therapy and subsequent TdP is higher in younger individuals and females.” (Please see Page 8, Lines 221–233)

  1. The discussion appropriately interprets the case report and literature review findings. It provides valuable insights into the risk factors and management strategies for QTc prolongation due to azole antifungals. The authors offer practical recommendations for clinicians, such as careful medication review and ECG monitoring. However, the discussion could be enhanced by suggesting specific protocols or guidelines for monitoring and management.

Response: We appreciate the reviewer’s comment. As the reviewer pointed out, it is desirable to specify parameters such as the exact timing and frequency of drug review and ECG monitoring. However, specific parameters for drug review and ECG monitoring related to QTc prolongation caused by azole antifungal agents have not been discussed in existing studies, nor can they be definitively stated based on the results of this review. After consulting with our co-author, a cardiologist, we have decided to avoid specifying these parameters to prevent potential misunderstandings among readers.

  1. The conclusion effectively summarizes the key findings and clinical implications. However, it could be strengthened by outlining specific areas for future research, particularly regarding the mechanisms of drug interactions and the development of safer antifungal therapies.

Response: We value the reviewer’s feedback. We have added sections regarding the mechanisms of drug interactions with antifungal agents and the ongoing research for the development of safer antifungal therapies to the main text. After careful consideration, we have decided to include these additions in the introduction and discussion section, rather than the conclusion, as it is more appropriate to reference the relevant literature in this context to aid reader understanding.

“These drugs may be involved in the elevation of plasma concentrations of QTc-prolonging drugs via the same metabolic pathway, owing to the inhibition of multiple cytochrome P450 enzymes in the liver and gastrointestinal tract [15–17].” (Please see Page 2, Lines 57–60)

“Furthermore, chemical modifications and the development of new formulations, including azole antifungal drugs, are being conducted to improve therapeutic efficacy and reduce adverse events. Nano-structured systems have been reported to act as carriers for antifungal drugs, potentially enhancing drug bioavailability and reducing toxicity [57]. These studies are expected to contribute to the reduction of adverse events associated with antifungal drugs, including azole antifungal drugs.” (Please see Page 11, Lines 378–383)

Reviewer 2 Report

Comments and Suggestions for Authors

This is a well-written, comprehensive, carefully vetted review of cases detailing the effects of anti-fungal azoles on the lengthening of the QTc portion of the EKG recording.  The authors carefully explain the consequences of this prolongation and introduce the resulting cardiac events that could occur. The concomitant lowering of potassium is intriguing as outlined in section 4.3.  They also mention other drugs and the level of effect they have on this phenomenon.  The discussion on confounding factors is good and recommendations given are especially important.  Raising awareness of the potential toxicity of azoles is a benefit of this review.

Major suggestion:

In Figure 2b in the case study, it would be helpful to microbiologists and others not trained in reading EKGs to use brackets and more arrows to indicate when the therapy started and when the QT interval lengthened and when TdP occurred in relation to the anti-fungal therapy. It also needs a separate figure to illustrate TdP to better explain the figure legend.

In the review portion, this reviewer had trouble reading the tables, in the supplement.  The tables are so large that it is difficult to interpret the case studies.  Perhaps taking the Tables and break them up, categorizing them by age or another important characteristic would be helpful.

As part of the review, are there any studies involving animal models to illustrate this toxic effect of azoles?  State this in the review – either identify this as a research gap or outline studies showing this toxicity.   Animal models have the advantage of excluding confounding factors, like other drugs and diseases.

Minor suggestion:

Please mention the range of a normal QT interval in humans in the Introduction so that the reader has an appreciation of the baseline before reading the aberrant QT intervals in the case study and in the review.

Author Response

Reviewer 2

This is a well-written, comprehensive, carefully vetted review of cases detailing the effects of anti-fungal azoles on the lengthening of the QTc portion of the EKG recording.  The authors carefully explain the consequences of this prolongation and introduce the resulting cardiac events that could occur. The concomitant lowering of potassium is intriguing as outlined in section 4.3.  They also mention other drugs and the level of effect they have on this phenomenon.  The discussion on confounding factors is good and recommendations given are especially important.  Raising awareness of the potential toxicity of azoles is a benefit of this review.

Major suggestion:

In Figure 2b in the case study, it would be helpful to microbiologists and others not trained in reading EKGs to use brackets and more arrows to indicate when the therapy started and when the QT interval lengthened and when TdP occurred in relation to the anti-fungal therapy. It also needs a separate figure to illustrate TdP to better explain the figure legend.

Response: We appreciate the reviewer’s comment. We have indicated the points at which QTc interval prolongation and TdP occurred using arrows in the figure 2. Additionally, to provide a clearer explanation of TdP, we have included an enlarged EKG of this case when TdP occurred, making it easier for readers to understand. Furthermore, regarding the definition of TdP, we have added the following to the annotation of figure 2.

“Torsade de pointes is a specific form of polymorphic ventricular tachycardia observed in patients with QTc prolongation. It is characterized by rapid and irregular QRS complexes that appear to twist around the baseline on the electrocardiogram.”

In the review portion, this reviewer had trouble reading the tables, in the supplement.  The tables are so large that it is difficult to interpret the case studies. Perhaps taking the Tables and break them up, categorizing them by age or another important characteristic would be helpful.

Response: We are grateful for the reviewer’s input. Following the reviewer's instructions, we have split the table in the supplementary material and made it more readable. Since this supplementary table provides a more detailed explanation of each case listed in table 1 of the main text, rearranging the case numbers or changing their order might make it harder to follow and potentially confuse readers. Therefore, we have deliberately kept the sequence and arrangement of the cases the same as in table 1 of the main text.

As part of the review, are there any studies involving animal models to illustrate this toxic effect of azoles?  State this in the review – either identify this as a research gap or outline studies showing this toxicity. Animal models have the advantage of excluding confounding factors, like other drugs and diseases.

Response: We appreciate the reviewer’s comment. We have added a mention of studies using animal models to investigate QTc prolongation caused by azole antifungal drugs in the main text of our paper.

“In animal experiments using long-eared white rabbits, it has also been shown that fluconazole may prolong the QTc interval and exert proarrhythmic activity by inhibiting human ether-a-go-go-related gene (hERG) protein trafficking [18].” (Please see Page 2, Lines 60–62)

Since our paper primarily reviews case reports of QTc prolongation caused by azole antifungal drugs in humans, including detailed findings from animal studies might confuse readers. Therefore, we have decided to limit the mention of studies using animal models to the main text and refrain from adding them to the review section to avoid any potential misunderstandings.

Minor suggestion:

Please mention the range of a normal QT interval in humans in the Introduction so that the reader has an appreciation of the baseline before reading the aberrant QT intervals in the case study and in the review.

Response: We thank you for the valuable suggestion, and we appreciate the reviewer’s comment. As per your suggestion, we have added a sentence regarding the normal range of the QTc interval in the introduction:

“QTc prolongation is described as a QTc interval extending beyond 450 ms in males and 470 ms in females, or an absolute increase of more than 60 ms from baseline [5].” (Please see Page 1, Lines 45– Page 2, Lines 47)